# Exploring Clinical Trials to Manage Firefighters’ Sleep Quality: A PRISMA Compliant Systematic Review

**DOI:** 10.3390/ijerph20053862

**Published:** 2023-02-21

**Authors:** Sara Alves, Josiana Vaz, Adília Fernandes

**Affiliations:** 1Instituto Politécnico de Bragança, Campus de Santa Apolónia, 5300-253 Bragança, Portugal; 2Institute of Health Sciences, Universidade Católica Portuguesa (UCP), 4169-005 Porto, Portugal; 3Centro de Investigação de Montanha (CIMO), Instituto Politécnico de Bragança, Campus de Santa Apolónia, 5300-253 Bragança, Portugal; 4Laboratório Associado para a Sustentabilidade e Tecnologia em Regiões de Montanha (SusTEC), Instituto Politécnico de Bragança, Campus de Santa Apolónia, 5300-253 Bragança, Portugal; 5Unidade de Investigação em Ciências da Saúde: Enfermagem (UICISA: E), Escola Superior de Saúde de Coimbra, 3000-232 Coimbra, Portugal

**Keywords:** circadian rhythms, occupational health, shift work, sleep

## Abstract

Sleep research has grown over the past decades and investigators are deeply involved in studying sleep and its impact on human health and body regulation. Despite the understanding that insufficient sleep is strongly linked to the development of several disorders, unsatisfactory sleep exposes health and safety to innumerous risks. The present study aims to review and analyze the main results of clinical trials, registered at ClinicalTrials.gov and ICTRT databases, and developed construct strategies to improve sleep quality on firefighters and enhance professionals’ sleep and health conditions. The protocol was registered in PROSPERO under number CRD42022334719. Trials registered between first registry and 2022 were included. We retrieved 11 registered clinical trials; seven met eligibility criteria and were included in the review. A relation between sleep disorders, shift work, and occupational health problems was found, and retrieved trials showed that sleep education programs can improve sleep quality and sleep hygiene. Science has already recognized sleep’s importance for metabolic functioning and survivorship. Nevertheless, it continues to play a major role in discovering methods to diminish the problems faced. Strategies contemplating sleep education, or intervention, should be presented to fire services to tackle this problem and promote healthier and safer environments.

## 1. Introduction

Quality of sleep is a complex health concept involving multiple and distinct characteristics. Genetics, physical and psychological conditions, family, and social environment are some of the factors which can influence sleep quality [1]. Besides having a difficult definition, the National Institutes of Health (U.S. Department of Health & Human Services) considers sleep quality to be important for human survival as food and water [2]. Recognized essential for one’s health, contentment, and functional capacity [3,4], sleep is connected to basic body functions, such as glucose metabolism [5], hormone regulation [6], and nervous and immune system control [7]. Sleep disruptions can cause a state of asynchronisms between the homeostasis of the circadian rhythm and the sleep–wake cycle, known as circadian misalignment [8,9,10]. Therefore, in the long term, sleep disturbance habits, and respective circadian misalignment, increase the risk of developing a range of negative health outcomes and chronic diseases [1,11,12]. Apart from harming health and wellbeing, evidence implies that sleep can also be decisive for the maintenance of intellectual capability and occupational productivity [13,14]. Further studies indicated that poor quality of sleep is mainly related to working factors [15]. Shift and night work, for instance, are queries which challenge the biological and social cycles, representing an occupational health concern for the international scientific community [12,16,17]. Sleep deprivation is then pointed out as a source of driving accidents [18] and work injuries [19,20,21,22]. Thus, sleep disruptions can represent a major issue in the working environment, being more noticeable in specific activities, such as health and public services [13]. Firefighting is one of the professions requested to provide support continuously and work on shifts, sometimes reaching, or working above, 24 h periods. When “on duty”, these professionals tend to rest in a different environment, usually prompt to abrupt awakenings, due to unpredictable emergency requests [23,24]. Overnight calls and requests influences firefighters’ sleep quality [25,26]. Nevertheless, poor quality of sleep in this population can also be associated with other work-related risk factors, such as musculoskeletal disorders and stress [27]. Underestimating these professionals’ quality of sleep can trigger the development of insomnia, restless legs, and shift work sleep disorders (SWD) [28]. It is necessary to know the different types of approaches that are effective in supporting the identification of this problem [29]. The development of more research to investigate different interventions, aimed at ensuring sleep quality for professionals who work in shifts, is also essential [30]. The present study aims to add to this subject by reviewing the main results of clinical trials realized to improve sleep quality in firefighting workers, following analyzing methods and programs developed to enhance professionals’ sleep and health conditions; it is also possible to verify if there is any gap in knowledge that could lead to further investigation of the subject.

## 2. Material and Methods

The present systematic review of clinical trials was elaborated following the PRISMA-S extension [31] and the flow diagram of PRISMA—Preferred Reporting Items for Systematic reviews and Meta-Analyses [32] as a guide for research; these provide more substantial instructions during the information retrieval process, for distinct types of data source and search methods [31]. Design and methods used comply with Centre for Reviews and Dissemination (CRD) Guidance for undertaking reviews in healthcare. The protocol of this systematic review was registered with PROSPERO (registration #: CRD42022334719).

### 2.1. Eligibility Criteria, Information Sources, and Search Strategy

The present review considered eligible original studies investigating programs and strategies realized to improve the quality of sleep on firefighters, that comprised participants above 18 or more years, with no time lime limitations and restricted to English, Spanish, French, and Portuguese. As per Table 1, which provides information regarding the eligibility criteria, developed following PICOS methodology [32], papers included focus on firefighters as participants, with no individual characteristic constraints and studies’ interventions aimed to enhance quality of sleep and overall health of firefighters, compared to usual care or no intervention. Trials still in the recruiting stage or not yet finished, and therefore with no results, were also accepted for protocol and prospective outcomes analysis. No other specific restrictions for the selection of the studies were applied. Unregistered clinical trials, studies in which interventions targeted other professionals or organizations apart from firefighting departments were excluded.

Information retrieval was realized independently by two researchers (SA and JV) utilizing the ClinicalTrials database and the WHO International Clinical Trials Registry Platform (ICTRP). These are the best well-known platforms for registry of clinical studies, being regulated by the U.S. National Library of Medicine [33] and World Health Organization, ensuring transparency and accountability of investigations [34] and making public research protocols and related outcomes [34].

The focus of the present work was to analyze clinical trials investigating interventions on firefighters’ sleep quality, performed around the globe, from first registry to July 2022. The first search was conducted in July 2022. Registrations were explored again on September 2022 to identify additional studies and new results or updates from the previous analyzed papers. The search strategy involves combining the following search terms, adapted for each registry:

ClinicalTrials.gov: “sleep wake disorders” OR “sleep disorder” OR “sleep problems” OR “sleep conditions” OR “Parasomnias” AND “firefighters” OR “fire fighter” OR “fire man”.ICTRP: (((“sleep problems”) OR ((“sleep”) AND (“problems”)) OR ((“sleep disturbances”) OR ((“sleep”) AND (“disturbances”)) OR ((“sleep conditions”) OR ((“sleep” AND (“conditions”))) AND (((“firefighters”) OR ((“fire”) AND (“fighters”))).

### 2.2. Study and Data Selection Process and Quality Assessment

The present search and data selection was conducted by two independent reviewers (S.A. and J.V.) through selection of scientific databases and evaluation of eligibility of clinical trials, starting by analyzing title and abstract. Afterwards, full text papers were screened by S.A. and reviewed by J.V. to verify if studies met the inclusion criteria. If discrepancies between reviewers occurred regarding eligibility, inclusion, or not, of studies, these were solved through consultation of a third independent reviewer (A.F.) to achieve agreement. Records were managed through Mendeley, specific software for managing bibliographies.

Data extraction was realized by S.A. and subsequently proofed by J.V. For each study, an electronic table containing the following topics was fulfilled: metadata, study characteristics, and participants info (Table 2).

Considering the small number of studies and the heterogeneity in the evaluation criteria, a narrative synthesis was carried out as part of the results of the present review.

The methodology quality of included articles was assessed, making use of the latest version of RoB 2: Cochrane risk-of-bias tool for randomized trials, which is the most recommended tool for any type of randomized studies, and the NOS (Newcastle–Ottawa quality assessment scale), as it can be applied for case–control and longitudinal studies, being the most frequently used tool by operators [35].

The Cochrane tool was developed to easily assess the risk of bias of randomized trials through the evaluation of a reported effect. The outcome is evaluated by comparing two interventions—referred to as experimental and control—for a specific result [36]. Every study using RoB 2: Cochrane risk-of-bias tool for randomized trials was rated, making use of the RoB 2 Excel tool. This instrument is divided into five bias domains: bias arising from the randomization process, bias due to deviations from intended interventions, bias due to missing outcome data, bias in measurement of the outcome, and bias in selection of the reported result. Once RoB2 is developed to assess bias for a specific result, each field has questions which allows reviewers to perform interrogation and make the decision as to whether there is bias or not [36].

The NOS (Newcastle–Ottawa Scale) is a modified validated tool from the original Newcastle-Ottawa instrument by the University of Pennsylvania Health System Center for Evidence-based Practice. A “star system” was developed by authors to assess and judge a study in three different areas: the selection of the study groups; the comparability of the groups; and the ascertainment of either the exposure or outcome of interest for case–control or cohort studies, respectively [37]. Studies in the review assessed using the Newcastle–Ottawa Scale ware rated using a number of stars in three domains and considered as poor (selection, ≤1; compatibility, 0; outcome, ≤1), fair (selection, 2; comparability, 1–2; outcome, 2–3), or good (selection, 3–4; comparability, 1–2; outcome, 2–3). The general description and an overall view of both assessment risk of bias of included trials in the review are shown in Figure 1 and Table 3.

## 3. Results and Discussion

To the best of our knowledge, this is the first systematic review to assess ClinicalTrials by reviewing its main results and corresponding assumptions. These studies plan to develop health programs or interventions, and the main objective is to improve sleep circumstances and habits of firefighters and, as a result, improve general health and conditions for a better performance at work.

### 3.1. Search Results

From the research conducted between July 2022 and September 2022, a total of 11 registered clinical trials were retrieved from the database ClinicalTrials.gov and ICTRP.

Following rejecting two duplicates, nine studies were assessed for eligibility. After analysis of full-text and study sample and phase, seven met the eligibility criteria and were included in the review. The flow diagram of retrieval trials is shown in Figure 2. Some authors discourage the elaboration of research works based only on registered clinical trials [40], as these are not yet correctly recorded on correspondent data bases. As such, we also developed a major exploration, including bibliographic databases such as MEDLINE and PubMed. The same combination of keywords was applied; however, we could not find further clinical trials to add to our study, which subsidizes such limitation.

Presented in Table 4 is a general description of the included trials considering registration year, country of development, study phase, number of participants, study design, type of intervention, and a description of principal goals and outcomes. Analyzing the progress phase of each trial, we could identify that four (50%) of these were completed, the remaining being either during recruitment stage, in a primary phase, meaning not recruiting, or during final evaluations to establish primary outcome measures. Linked publications were found for three of the four completed trials, and can be obtained, as a journal paper, in “Sleep”, an official journal of the Sleep Research Society, entitled “Randomized, Prospective Study of the Impact of a Sleep Health Program on Firefighter Injury and Disability” [41], “Sleep Medicine”, a journal of the World Sleep Society and International Pediatric Sleep Association, designated “Sleep quality and sleep disturbances among volunteer and professional French firefighters: FIRESLEEP study” [42], and “ Effect of Zolpidem on Sleep Quality of Professional Firefighters; a Double Blind, Randomized, Placebo-Controlled Crossover Clinical Trial” at Acta Medica Iranica, a Tehran University of Medical Sciences journal. One of the completed trials reported results at ClinicalTrials.gov [42]. Ongoing active studies were started in 2021, aiming to analyze 2400 participants by completion time.

The included seven studies involved a total of 4489 individuals (12–2400), with a mean of 561 and median of 112.5 participants per study, most participants being from USA (4276 individuals, 95.3%). Following the same pattern, most clinical trials were conducted in the USA, and the remaining studies in France and Iran, distributed between 2011 and 2021 (Figure 1).

All participants have ages above 18, but only three of the completed trials delivered information regarding gender specifications, the large majority being male (96%). We found that all of the participants had the same type of characteristics, and performed similar functions; however, queries regarding their cultural and occupational differences were raised in some studies, as per the study of Stefani (2011), which did not include women. In order to facilitate the understanding of the trials’ analysis, these were divided taking into account the type of intervention programs designed: sleep education trials and sleep crossover and observation trials, followed by a discussion regarding the scientific awareness potential of the subject, and finalizing with the quality assessment.

### 3.2. Sleep Education Trials

Most clinical studies (75%) intended to develop interventional programs, mainly through the application of sleep education programs. Barger (2021) with the SHEP (Sleep Health Education Program) [46], and Green & Christopher (2021) with CBTI-F (Cognitive Behavioral Therapy—Insomnia) [43] both developed randomized open labeled trials programs. The SHEEP program [46], with an estimated enrollment of 2400 participants, was designed to deliver an early sleep health education program to a randomized experimental group. Investigators propose to verify if enrolled firefighters had improved their health and safety outcomes (alterations of injuries, disability days, and vehicle crashes numbers), compared to the controlled group. The CBTI-F program [43] aims to deliver psychoeducation on sleep, and therapy for insomnia, to a randomized group of 23 firefighters in total. The treatment group will be part of a training program, whereas the control group will only be recommended the training at the end of the study. This trial aims at diminishing occupations concerns in the training group by promoting the adoption of actions conducive to good quality of sleep. Although with inconclusive results, as both programs are not yet finalized, if competent, they will prove that education interventions are able to improve health and safety of firefighters and ameliorate quality of sleep and general health plus decrease levels of stress and anxiety and increase levels of vigilant attention, bettering occupation health conditions.

The Firefighter Fatigue Management Program: Operation Fight Fatigue [44], and the Comprehensive Firefighter Fatigue Management Program (CFFMP) [42], both from principal investigator Czeiler (2013), are completed interventional sleep education and screening programs regarding fatigue-related hazards and sleep hygiene. The “Operation Fight Fatigue” [44] enrolled 620 firefighters in a web-based education session concerning basic sleep hygiene. This study applied instruments to assess sleep-related fatigue concerns on professionals’ safety, health, and performance. Later, they conducted a sleep optimization program, within the intervention group, to promote strategies to maximize sleep and ease fatigue. Contemplating 1189 participants, the main goal of CFFMP [42] was to demonstrate that the application of effective programs, such as “Operation Healthy Sleep”, could diminish work injuries, absence, and accidents and improve job satisfaction and ability to cope with associated stress factors in the interventional group, while the control group will continue its normal duties. The participative group was invited to attend education presentations focusing on firefighter mortality, fatigue-related incidents, and to discuss strategies to promote alertness. The group was also encouraged to fulfill a self-reported tool, to identify individuals with high risk of sleep disorders, and be referred to sleep clinics for follow-up.

Previous studies, in different populations, suggest that it is still unclear if sleep educational programs produce better sleep hygiene behaviors and, thus, a better quality of sleep [47]. It was also found in the literature that sleep knowledge is improved through sleep education interventions, meaning that programs focused on sleep education are effective at cutting inadequate sleeping habits and making lifestyle changes [48]. Nevertheless, it is not accompanied by sleep quality enhancements or changes in sleeping patterns [49]. These findings corroborate the results from CFFMP [42], which concludes that after the sleep education intervention, attending participants (45.6%) reported a higher total sleeping time, lower number of times falling asleep in meetings or over the telephone, lower incidence of sleeping while driving/stopped in traffic, and fewer vehicle crashes and job injuries due to exhaustion. The CFFMP intervention [42] informs that education programs do have positive outcomes such as improvement of alertness and performance, diminishing occupational hazards, and increasing general health indices. However, this intervention does not assess if there were changes in sleep quality, focusing mainly on fatigue-related health risks and tactics to stimulate alertness. Adding to this, the Firefighter Fatigue Management Program: Operation Fight Fatigue [44] does not present final outcomes, inhibiting us to analyze results and further discussion. As such, the question of whether sleep education programs are effective at improving sleep quality remains open. Hopefully, the ongoing Sleep Health Education Program [46] and the CBIT-F trial [43] will be able to add to this question and inform the scientific community for the impact of spreading knowledge regarding sleep and sleep hygiene to target changes in sleep habits, promoting sleep quality, health, and better quality of life.

### 3.3. Sleep Crossover and Observation Trials

An Iranian study from principal investigator Amir Stefani (2011) [39], whose interventional method differs from the mentioned educational studies, opted for therapeutical administration strategy of medication vs. placebo. With a sample of 20 participants, the study divided the sample in two groups: the intervention group, where Zolpidem was administered at bedtime for 14 days as intervention, and the placebo group, having wheat flour tablets at bedtime for 14 days as control. This crossover study concludes that the administration of this non-benzodiazepine can improve firefighters’ sleep and sleep conditions in general. Despite these results, recent publications and guidelines highlight its addictive potential, misuse, and overprescription being linked to more adverse events than benefits. Daytime sleepiness, memory problems, road accidents, and, consequently, deaths are pointed out as side effects associated to Zolpidem [50,51], leading to the conclusion that utilizing Zolpidem should only be considered as a last-resort intervention to improve sleep problems due to possible medication-usage-related concerns and already existing occupation health risks and sleep difficulties in firefighters [50,52]. Adding to this, and despite the small evidence produced, researchers found that firefighters tend to turn to substance abuse and alcoholism as a short-term way to cope with occupational struggles [53]. As such, utilizing substances in a population prone to overusing is not advisable. The outcome of this study highlights that medication should only be used as part of a controlled treatment program, during short time periods, to control sleep problems/pathologies.

The Circadian Rhythm Disruption Effects on Smoke Inhalation study [45] from principal investigator Quindry (2021) is still ongoing. They aim to develop an interventional controlled study, restricting 15 participants’ sleeping time and then exposing them to woodsmoke, in a lab-controlled environment. Their intent is to assess the effects of sleep deprivation and circadian rhythm disruption on the incited inflammatory response, post smoke exposure, which represents a major risk for firefighters. Previous investigations already showed that the circadian clock is linked to the regulation of organism’s immune responses in regular conditions [54,55,56] and is under the control of suprachiasmatic nucleus of hypothalamus [57]. However, when an infection occurs, the circadian regulation turns its focus to the reorganization of immune responses, in order to be able to fight the source of the disorder [58]. Findings suggest that factors which can disturb the circadian rhythm are irregular eating patterns, sleep/wake time, and night shift. When these circumstances occur repeatedly, they can stealthily increase inflammation [59] by disrupting central clock components [60] and culminate in circadian-rhythm-associated diseases [59]. It is also a well-known fact that wood smoke, regardless of its origin, produces a wide range of gases and particles hazardous to health, damaging the airways, which can cause inflammation [61] and the reduction of immune cells levels in lungs, suggesting that these elements may be cytotoxic [62].

A particular investigation on firefighters noted that, following smoke exposure, an increase of circulating white blood cells and PMN counts occurred. This outcome provides some indication that smoke inhalation provokes a systemic inflammatory response from the organism, which can culminate in exacerbations of existing respiratory problems and precipitate cardiovascular events [56].

Although without final results from Quindry’s trial, evidence suggests that the combination of inflammatory factors and circadian misalignment will possibly exponentially increase inflammatory markers, which can, in the long term, raise the incidence of occupational-related disorders and incapacity [63,64,65,66].

The completed French study, entitled FIRESLEEP [38], is a prospective quality assessment research, focused on investigating the sleep quality of professional and volunteer firefighters. It also aims to measure excessive daytime sleepiness prevalence in this population and point out possible risk factors linked to bad quality of sleep. The various validated questionnaires, applied to 196 participants, revealed that 6.9% of the firefighters had poor sleep quality, 27.7% showed excessive daytime sleepiness, 18.8% reported moderate to severe symptoms of insomnia, and 1.6% had moderate to high risk of obstructive sleep apnea. These results confirm the overall literature [67,68,69,70,71] showing that poor sleep quality and sleep disturbances are present in the firefighting work class. The Firesleep program was sufficient at establishing the association between certain risk factors and quality of sleep. Outcomes proved the association between anxiety and poor sleep quality and also that excessive daytime sleepiness and insomnia are linked to bad quality of sleep [72]. Apart from presenting alarming results, the implemented medical history questionnaire did not ask questions about signs of post-traumatic stress disorder or burnout. As such, the assumption that the number of night calls is an independent risk factor for bad sleep is not entirely accurate [28]. Findings from surveys and questionnaires are indubitably important in research [72]. These instruments allow us to increase communication about the quality of sleep of firefights. It would be very interesting, parallel to educational sessions and sleep clinic attendance, to analyze if such actions have had a positive impact on performance. If investigators ally experiments with real results, it is possible to revolutionize work schedules and habits to promote healthier workers.

### 3.4. Scientific Awareness Potential

The impact of occupational health and its relation to sleep quality is expanding globally, and its impact on health has become a focus both for scientists and employers [73,74]. During this study, research through the PubMed database was developed, following the same search strategy utilized for the identified clinical trials. It was observed that scientific investigation on firefighters’ sleep has gradually grown since the very first related published paper, which was reported in 1994. Since that date, until 2022, 206 papers can be found, with a substantial growth of research and publication in 2021 being noticeable, when 20% of the articles were released. Worley, in 2018, described that the expansion in research among sleep subjects had been evident since 2005, the year in which there was a significant increment in the number of sleep journals [75]. In addition, between the years of 2003 and 2012, scientific publications related to sleep more than doubled worldwide, strongly contributing to communicating findings about sleep functions and emphasizing its impact on health, safety, and quality of life [76].

Firefighting belongs to one of the activities raising more concerns among the scientific community due to the dangerous tasks and hazards these professionals encounter [26].

Chronologically, publications regarding firefighters’ health conditions substantially expand from 2011, a period which corresponds to 10 years after the World Trade Center disaster (WTC). Further research indicates that, from this point to recent days, longitudinal studies located in USA visibly started to expand, focusing on firefighters’ occupational health and exposure disease risk, resulting in more than 40 studies. WTC-related exposure disease findings may explain the reason why most of the registered clinical trials found are realized in USA, while other countries do not invest as much in this matter. In the past year of 2021, it is possible to notice another boost of research, this time extended to European countries, for example, Spain, Poland, and Germany, and other countries such as Turkey and Cyprus, as during the pandemic time, the world did become more aware of the occupational challenges the health systems face [77]. This scenario could be the origin of more investigations, as firefighters were applauded and recognized as essential for public safety and have continued to work regardless of the challenges of COVID-19 [78]. The increase in studies in this area shows that the scientific community is aware of and alerted to this problem. The results of science can serve as foundations for the implementation of government strategies to safeguard these professionals.

### 3.5. Quality Assessment

The quality evaluation of integrated studies is found in Figure 1 (ROB2 quality assessment) and Table 3 (Newcastle–Ottawa Scale assessment of study quality). It was found that most protocols feature some bias concerns; these do not describe in total the methodology to be used during the randomization process, and some do lack explanation of outcome measurement and results. All selected studies use randomization, with only one study following double-blind randomization. For the remaining studies it is not completely explicit how this process was developed. Furthermore, information regarding missing outcome data and assessment of outcome and reported results is not presented, as studies are still ongoing, facilitating risk of attrition and reporting bias.

In addition, the use of only male participants in two of the selected studies could potentially lead to a conceptual and methodological gender bias with partially invalid results that affects women’s health [79]. As so, more wide-ranging research, extended to other countries and including both genders (male and female), is needed. It would be just as important to understand the work and social specifications of that group to target the origin of sleep disturbances and apply accurate educational and interventional programs. Adding to this question, four studies (Green & Christopher (2021), Quindry (2021), and Estephani (2011)) reported small sample sizes, which limits statistical significance of results and decreases the representativeness of the population, raising the risk of potential bias.

Only completed trials from authors Czeiler (2012) and Lockey (2013) show low risk of bias, since these describe in detail all steps of the respective studies, from the randomization process to missing data and reported results.

Thus, although there are studies with a lack of methodological information, these are mainly due to the fact that they are still ongoing. Since investigations are registered in international databases, and the confidence level and risk of bias are established.

## 4. Conclusions

The present study contributes to the understanding of sleep quality and its influence on the occupational health and general wellbeing of firefighters.

The analyzed ClinicalTrials allowed for a relatively better awareness of the low sleep quality of firefighters. Although scientific evidence has shown, throughout the years, that bad conditions of sleep lead to serious health problems such as exhaustion, failure to focus, and action–reaction incapability, the reduced number of clinical trials for interventions highlights the necessity of more research to be placed into plan. Adverse work-related circumstances can be particularly severe for firefighters, who, when placed in a dangerous situation, need to be at full mental and physical capacity for immediate response, to properly execute actions and implement strategies. Occupational health programs contemplating sleep education, or intervention, should be presented to fire services to tackle this problem and ensure a more active role in promoting healthier and safer work environments.

It would be also important, for further studies, to move beyond surveys and assumptions. Implementing action plans, whether educational or interventional, and evaluating if such measures do cause positive impact on firefighters’ quality of sleep are needed.

## Data Availability

Not applicable.

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
