# Peer review of "Exploring Clinical Trials to Manage Firefighters’ Sleep Quality: A PRISMA Compliant Systematic Review"

_ijerph, 2023, doi:10.3390/ijerph20053862_

Round 1

Reviewer 1 Report

In general, the manuscript proposes a systematic review of an interesting topic that has gained relevance in recent years. It is well written and presented in a friendly form.

The following are the comments and suggestions for the authors:

1. Sometimes the citation looks like  APA and others like IEEE or another standard, for example this case (Barger at al., 2015). 

2. Figure 2: It has a column named intention-to-treat without content in the rows. Table 4 contains red underline on third row. The size of this figure and table would be improved to fit better in the manuscript.

3.I consider that more information related to results shown in Figure 2 and Table 4 should be given. For example why studies in the table are not in figure? In addition, more information about what the parameters in Table 4 mean: Representativeness of exposed cohort, Selection of non-exposed cohort, ......

4. Check the following TYPOS:

a. the Fire Fighter Fatigue Management Pro

b. ‘safety, health, and performance (quotation without closing)

c. the study dived the sample

d. Sometimes, including the title, authors open quotation marks but these are not closed. Additionally, single and double quotation are used indistinctly.

Author Response

Response to Reviewer 1 
In general, the manuscript proposes a systematic review of an interesting topic that has gained 
relevance in recent years. It is well written and presented in a friendly form.
The following are the comments and suggestions for the authors:

1: Sometimes the citation looks like APA and others like IEEE or another standard, for example 
this case (Barger at al., 2015). 
Authors response: Thank you for your comments. As suggested by the reviewer we have revised 
all citations and corrected them according to the IJERPH's recomendations and guidelines.

2. Figure 2: It has a column named intention-to-treat without content in the rows. Table 4 
contains red underline on third row. The size of this figure and table would be improved to fit 
better in the manuscript.
Authors response: Authors apologize for this incident and thank you for noting it. Figure 2 is 
related to the "ROB2 quality assessment", which was retrieved directly from the ROB2 assessment 
tool. The refered column "intention-to-treat" was removed, as it is not applicable to the type of review 
presented. Table 4 was corrected and both figure and table changed to better fit the manuscript, as 
suggested.

3. I consider that more information related to results shown in Figure 2 and Table 4 should be 
given. For example why studies in the table are not in figure? In addition, more information 
about what the parameters in Table 4 mean: Representativeness of exposed cohort, Selection of 
non-exposed cohort, ......
Authors response: Authors thank the reviewer for raising this concern . Studies in table 4 are not in figure 2 as they refere to different study metodologies. Studies in table 4 were assessed following 
the "Newcastle-Ottawa scale assessment" as these correspond to case-control and longitudinal studies. Studies in figure 2 refere to randomized trials, and therefore were accessed following RoB 2: Cochrane risk-of-bias tool (these are explained and sinalized in table 1: General description of 
included trials). Please note that due to proposed alteration, table 4 is now table 3 and figure 2 is now identified as figure 1.

4. Check the following TYPOS:
a. the Fire Fighter Fatigue Management Pro
b. ‘safety, health, and performance (quotation without closing)
c. the study dived the sample
d. Sometimes, including the title, authors open quotation marks but these are not closed. 
Additionally, single and double quotation are used indistinctly.
Authors response: The authors thank the reviewer for noticing the typo and apolagise for such 
fault. We have double checked and corrected any grammatical inconsistency.

Reviewer 2 Report

Dear authors, you made a systematic review investigating the sleep of firefighters and interventions, as those being professionals with a high impact on their sleep quality and the consequences of bad sleep. the review was structured on the PRISMA model, and registered at Prospero, enhancing the quality of the methodology. 

The search strategy was performed on 2 data sets for clinical trials. This limitation might have reduced the number of studies, as only more recently the registration of trials is more commonly spread. Maybe there should be included a platform like PubMed, Embase etc., or at least include a paragraph in the discussion that the limitation of the search strategy to platforms for registration of clinical trials might have limited the number of studies.

The search strategies use correct keywords/mesh terms, allowing a very inclusive search strategy, results for each step of the review was shown in the flow-chart.

The evaluation of the studies was based on standard tools (Risk of bias-Rob, and the Newcastle Ottawa).

Results are well exposed, tables are clear and complete.

Discussion and conclusion in the agreement with the objectives and the methods.

Author Response

Dear authors, you made a systematic review investigating the sleep of firefighters and interventions, as those being professionals with a high impact on their sleep quality and the consequences of bad sleep. the review was structured on the PRISMA model, and registered at Prospero, enhancing the quality of the methodology.
Authors response: Dear reviewer, we do appreciate your work and your praises to our manuscript.

1. The search strategy was performed on 2 data sets for clinical trials. This limitation might have reduced the number of studies, as only more recently the registration of trials is more commonly spread. Maybe there should be included a platform like PubMed, Embase etc., or at least include a paragraph in the discussion that the limitation of the search strategy to 
platforms for registration of clinical trials might have limited the number of studies.
Authors response: The authors thanks the reviwerer for raising an interesting point regarding the limitation of the search strategy to clinical trials registration platforms. We do understand that some authors discourage the elaboration of research work based only on registered clinical trials, as these are not yet correctly recorded on correspondent data bases. As so, we have developed a major exploration, including bibliographic databases, as MEDLINE and PubMed, utilizing the same combination of keywords to enrich the discussion and the reached conclusions, as well as to ensure that that important works were not missed from the review. Once the occupational exposure of firefighters is still a recent subject of interest, no other studies were found in broader platforms. The authors included a paragraph discussing this limitation as suggested by the reviewer.

2. The search strategies use correct keywords/mesh terms, allowing a very inclusive search 
strategy, results for each step of the review was shown in the flow-chart.
Authors response: The authors really appreciate your notes on the manuscript. 

3. The evaluation of the studies was based on standard tools (Risk of bias-Rob, and the  Newcastle Ottawa).
Authors response: The authors are grateful for you evaluation. 

4. Results are well exposed; tables are clear and complete.
Authors response: The authors are very pleased for you coments. 

5. Discussion and conclusion in the agreement with the objectives and the methods.
Authors response: We do appreciate your work and your praises to our manuscript.

Reviewer 3 Report

Add the aim of the study in the abstract and the end of the introduction.

Line 45 there is a space 

Materials and methods ... after results!

Author Response

1. Add the aim of the study in the abstract and the end of the introduction.
Authors response: Authors thank the reviewer for raising this concern. Both abstract and intruduction were corrected and rewritten for a better explanation of our aim in the present study. 

2. Line 45 there is a space.
Authors response: Thank you for noticing this error. It was emended.

3. Materials and methods ... after results!
Authors response: Thank you for your comments. The authors agree with the reviewer's comment regarding the position of the “Materials and Melhods” chapters, and made the correct modifications.

Round 2

Reviewer 3 Report

Thank You for the bog effort in improving your paper.

I accept to in the present form.